

# Hurricane María drives increased indoor proliferation of filamentous fungi in San Juan, Puerto Rico: a two-year culture-based approach

Lorraine N. Vélez-Torres[1], Benjamín Bolaños-Rosero[1], Filipa Godoy-Vitorino[1], Felix E. Rivera-Mariani[2], Juan P. Maestre[3], Kerry Kinney[3] and Humberto Cavallin[4]

[1] Department of Microbiology & Medical Zoology, University of Puerto Rico, Medical Sciences Campus, San Juan, PR, USA
[2] College of Biomedical Sciences, Larkin University, Miami, FL, USA
[3] Department of Civil, Architectural, and Environmental Engineering, The University of Texas at Austin, Austin, TX, USA
[4] School of Architecture, University of Puerto Rico, Río Piedras Campus, San Juan, PR, USA

Corresponding author
Benjamín Bolaños-Rosero,
benjamin.bolanos@Upr.edu

## ABSTRACT

Extensive flooding caused by Hurricane María in Puerto Rico (PR) created favorable conditions for indoor growth of filamentous fungi. These conditions represent a public health concern as contamination by environmental fungi is associated with a higher prevalence of inflammatory respiratory conditions. This work compares culturable fungal spore communities present in homes that sustained water damage after Hurricane María to those present in dry, non-flooded homes. We collected air samples from 50 houses in a neighborhood in San Juan, PR, 12 and 22 months after Hurricane María. Self-reported data was used to classify the homes as flooded, water-damage or dry non-flooded. Fungi abundances, composition and diversity were analyzed by culturing on two media. Our results showed no significant differences in indoor fungal concentrations (CFU/m$^3$) one year after the Hurricane in both culture media studied (MEA and G25N). During the second sampling period fungal levels were 2.7 times higher in previously flooded homes (Median = 758) when compared to dry homes (Median = 283), ($p$-value < 0.005). Fungal profiles showed enrichment of *Aspergillus* species inside flooded homes compared to outdoor samples during the first sampling period (FDR-adjusted $p$-value = 0.05). In contrast, 22 months after the storm, indoor fungal composition consisted primarily of non-sporulated fungi, most likely basidiospores, which are characteristic of the outdoor air in PR. Together, this data highlights that homes that suffered water damage not only have higher indoor proliferation of filamentous fungi, but their indoor fungal populations change over time following the Hurricane. Ultimately, after nearly two years, indoor and outdoor fungal communities converged in this sample of naturally ventilated homes.

# INTRODUCTION

Fungi are ubiquitous eukaryotic organisms of environmental and medical importance. Exposure to airborne fungi is associated with adverse health effects in humans (*Kendrick, 2011*; *Cannon et al., 2018*). Fungi reproduce by spore production, many of which are dispersed in the air. Size of these spores ranges approximately 0.65 μm to >20 μm (*Portnoy, Barnes & Kennedy, 2008*; *Hamilos, 2010*; *Li et al., 2011*; *Kwon-Chung & Sugui, 2013*; *Claub, 2015*). Fungal spores and fragments less than 2.5 m can deposit in the alveoli and trigger asthma in sensitized individuals (*Joubert et al., 2020*). In contrast, large fungal spores and clusters of small fungal spores can deposit in the upper airway triggering allergic rhinitis (*Joubert et al., 2020*). Environmental fungal contamination, including high airborne fungal spore concentrations in the air, are associated with inflammatory respiratory conditions such as asthma and allergic rhinitis (*Lewis et al., 2019*). In addition, for people already suffering from asthma, exposure to fungal spores simultaneously with other allergens (animal dander, dust mites, cockroaches) may cause dangerous exacerbations of existing asthma (*Gautier & Charpin, 2017*). Approximately, 339 to 500 million people worldwide are affected by asthma and allergic rhinitis, respectively (*Kołodziejczyk & Bozek, 2016*; *Global Asthma Network, 2018*). In Puerto Rico (PR), asthma prevalence in children and adults is higher than in the United States (2017: 15.5% *vs* 7.5% for children and 12.2% *vs.* 9.1% for adults) (*Ortiz-Rivera, 2018*) .

Due to its tropical location and frequent rain events, PR has a high concentration of fungal spores in the air. Very high levels observed in PR exceed 110,000 spores/m$^3$, 2.2 times higher than those observed in the other National Allergy Bureau (NAB) of the American Academy of Allergy Asthma and Immunology mold levels stations (*AAAAI, 2021*). Outdoor fungal spore concentrations typically reach their maximum levels in PR during the rainy months of September to November (*Quintero, Rivera-Mariani & Bolaños Rosero, 2010*). The fungi found in outdoor ambient air are typically split between Basidiomycetes (60%) and Ascomycetes (40%) (*Rivera-Mariani et al., 2020*; *AAAAI, 2021*). *Pleurotus ostreatus, Chlorophyllum molybdites* and *Ganoderma applanatum* are some of the basidiomycetes common in the air of PR (*Rivera-Mariani et al., 2011*). Ascomycetes include filamentous *Aspergillus* species (*A. fumigatus*, *A. terreus*, *A. niger*, *A. flavus*) which are the most frequent causative agents of human aspergillosis, lung inflammation and among the most frequent triggers of asthma and allergic rhinitis (*Portnoy, Barnes & Kennedy, 2008*; *Hamilos, 2010*; *Kwon-Chung & Sugui, 2013*). Indeed, a recent study in PR found that increases in outdoor fungal spore concentrations were associated with increases in asthma-related health claims, confirming that fungal spores are triggers of asthma (*Lewis et al., 2019*). Most Puerto Rican homes have natural air ventilation; therefore, spores present outdoors can easily penetrate indoors through open windows and doors. Even though the 2017 outdoor fungal spore season was lower than expected as a result of Hurricane María, the fungal spore season for

2018 was the highest ever recorded in the 16-year history of data collection at the San Juan station most likely due to the accumulation of organic debris in the aftermath of Hurricane María (*AAAAI, 2021*; *Bolaños-Rosero, 2021*).

Essential conditions for indoor growth of filamentous fungi include high moisture and humidity, warm temperature, poor ventilation or air circulation, and the presence of organic source materials (*Institute of Medicine, 2004*; *WHO, Regional Office for Europe, 2009*). Taking these factors into consideration, the extensive flooding caused by Hurricane María in PR had the potential to provide a favorable environment for fungal proliferation indoors in homes with water damage. Several studies have evaluated the effects of Hurricanes on airborne fungi in the United States, with the most common fungi isolated being *Aspergillus*, *Penicillium* and *Cladosporium* (*CDC, 2006*; *Chew et al., 2006*; *Rao et al., 2007*; *Schwab et al., 2007*; *Rabito et al., 2008*; *Saporta & Hurst, 2017*; *Gargano et al., 2018*; *Chow et al., 2019*; *Kontoyiannis et al., 2019*). Although PR has been affected by more than 15 Hurricanes and tropical storms since the 1980's, the effect of these natural disasters on the concentration and composition of indoor airborne fungi has not yet been studied. Here, we quantify airborne fungal concentrations in the months following Hurricane María in PR. Our hypothesis is that homes which sustained flooding or water damage will have higher fungal diversity and abundance of fungi related with damp indoor environments. In addition, since the sampled homes are predominately naturally ventilated, we hypothesize that fungal taxa in the indoor environment of dry non-flooded homes may reflect the taxa in the outdoor environment. We believe this study will provide the basis for understanding the effect of fungal growth and species composition as a result of a flooding event in PR. The results from this study lay the foundation to design mitigation strategies to minimize adverse fungal exposures after flooding events to better control lung inflammation and asthma in PR and other tropical countries.

## MATERIALS AND METHODS

### Study population

After obtaining Institutional Review Board (IRB) approval from the University of Puerto Rico - Río Piedras Campus (IRB protocol 1718-058), 50 residences from the area of Santurce (San Juan, PR) were selected to participate in this study (Fig. 1). Many homes located in this area sustained flooding during and after Hurricane María's passage through the island. Of the 50 homes, 10 were located in a non-flooded area and 40 were located in a flooded area according to a reference map generated by the Federal Emergency Management Agency (FEMA). In addition to considering this classification, participants were surveyed and asked to describe in more detail the level of water damage. This reported data allowed the classification of each household in greater detail: 13 houses were dry non-flooded, 24 houses had water-damage (received water from above, roof and/or windows), and 13 houses were flooded (received water from below, through doors and crevices). We acknowledge that the sample size for the different categories was small, but we were working with recruitment under very difficult circumstances. We thus have two types of classifications for each home: (1) FEMA based: whether the homes were located

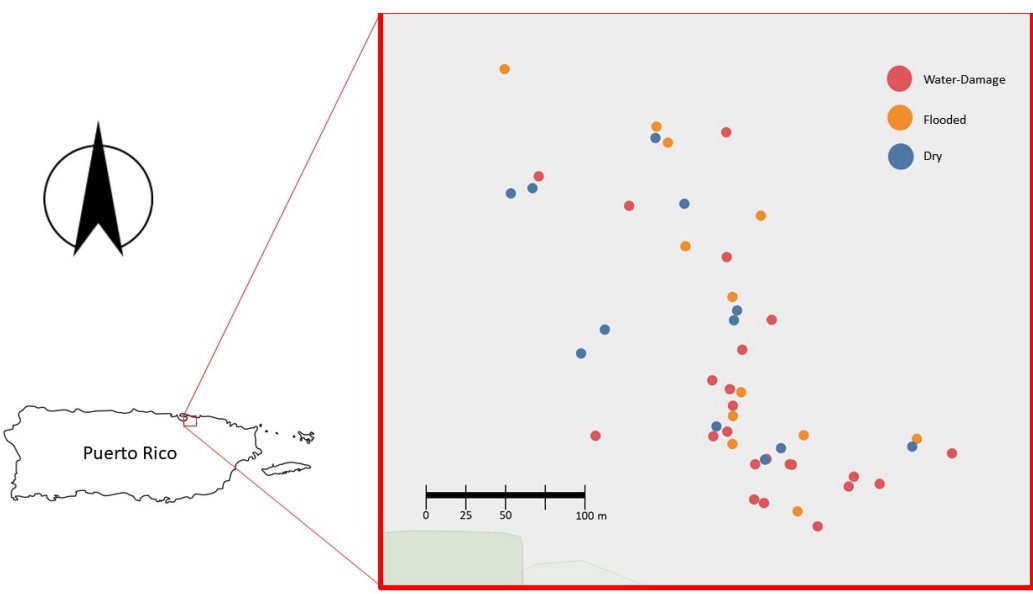

**Figure 1** **Map of the homes sampled in San Juan, Puerto Rico ($n = 50$).** Blue, red and yellow dots corresponds to Dry, Water-Damage and Flooded homes, respectively.

in a flooded or non-flooded area according to FEMA (dry area or flooded area) and (2) self-reported damage reported by study participants (dry, water-damage or flooded). After obtaining written informed consent, participants were asked to complete questionnaires to gather information on the demographics, health of home occupants (respiratory and psychological), home characteristics (ventilation, number of windows, construction materials) and the physical impact of Hurricane María on the home (water-damage, remediation activities, and the presence of visible mold). For the purpose of this study, we focus on evaluating the relationship between the self-reported degree of water damage and viable fungal communities present in samples collected from the indoor and outdoor environment of 50 homes in PR.

## Air sampling & fungal culturing

We collected indoor and outdoor samples during two different sampling periods: August–September 2018 and June–July 2019, 12 and 22 months, respectively, after Hurricane María. For the air sample collection, we used the MicroBio MB2 Bioaerosol Sampler (Cantium Scientific, Dartford, UK). This volumetric instrument collects airborne bioaerosol contaminants by aspirating air through a removable stainless-steel head (cover) which contains 220, one mm holes. This design creates a laminar airflow that impels the microorganisms, in this case the fungal spores, onto the surface of the culture media. A pilot study using different sampling volumes allowed us to set the optimal total air sampling volume at 60 liters (flow rate of 60L/35 s), which allowed us to count and identify to genus level the viable fungi without overgrowth on the Petri plate. Prior to Hurricane María, the standard sampling volume used in a previous study was 150 liters, but due to record-high outdoor fungal spores levels reported in 2018, the sampling volume was decreased to 60

liters (*Bolaños-Rosero et al., 2013*). At each home, indoor samples were collected at a height of 1 meter from four indoor locations (living room, kitchen, bedroom and bathroom) and an outdoor sample was collected in the front yard of the residences. Before sample collection, interior fans were turned off and the number of open and closed windows were counted. We used two types of culture media, Malt Extract Agar (MEA) and 25% Glycerol Nitrate Agar (G25N; $K_2HPO_4$ 0.375 g, Czapek concentrate 3.75 ml, yeast extract 1.75 g, glycerol 125 ml, agar 6 g, distilled water 375 ml). MEA is a nutritive growth media used for the isolation of general fungi. G25N is a selective growth media for xerophilic (dry tolerant) fungi, which grow at low water activity or availability (*Wu, Su & Ho, 2000*; *De León et al., 2018*). After sampling, Petri plates were incubated at $25 \pm 2$ °C for up to 2 weeks. Visible fungal colonies were counted using a stereomicroscope as early as 24–48 h after sampling with MEA, taking precautions to avoid mixing of the fungal colonies such as keeping the inverted plates horizontally. This process was repeated daily for 7 days in the case of MEA and for 14 days in the case of G25N. Culturable fungi were reported in colony forming units per cubic meter (CFU/m$^3$) using the MicroBio PC Reporter software and data from the fungal colony count with positive-hole correction and the total volume of air drawn through the MicroBio MB2 Bioaerosol sampler for each sample collected. Positive-hole correction accounts for the probability that more than one viable particle could be entering through a sampling hole, merging with other microorganisms at the same impaction site and producing a single colony (*Macher, 1989*). In addition to determining culturable fungi at each participating home, we also measured relative humidity levels both indoor and outdoor at each home. We also compiled precipitation data from the National Weather Service Forecast Office (NWSFO) and outdoor fungal spore levels from the San Juan AAAAI station for both sampling periods to characterize the ambient fungal background levels during the sampling events (*NWSFO, 2014*; *AAAAI, 2021*).

## Fungal identification

We performed slide mounts to analyze the colonies under the microscope and identify them to the genus level. First, we used the stereomicroscope to evaluate how many possible distinct fungi we had in our sample. Prior to preparing the slide mounts, we isolated the most predominant fungi present for further experiments. Then, we used fungi tape and lactophenol cotton blue staining to prepare the slide mounts. Following this, we observed the fungal slide mounts under the microscope and the fungi were identified to the genus level by evaluation of morphological characteristics such as spore shape, size, color and arrangement (*Samson et al., 2019*) .

## Statistical tests and data analyses

Data on fungal colonies concentrations and metadata variables were analyzed in RStudio©Version 1.2.1335 (R Foundation for Statistical Computing, Vienna, Austria) (*RStudio-Team, 2020*). When analyzing the indoor data, we used the value of the calculated median of the indoor measurements per home. To evaluate differences in indoor relative humidity by degree of water damage (dry, water-damage, flooded), we performed the non-parametric Kruskal Wallis test. We also used the non-parametric test Kruskal–Wallis

to analyze differences in fungal colonies concentrations as a function of different self-reported degree of water damage (dry, water-damage, flooded). We considered a *p*-value less than 0.05 statistically significant. To evaluate the contribution of outdoor fungal levels to indoor fungal levels, we performed a Spearman Correlation analysis and a Linear Regression analysis with RStudio for graphs and Intellectus Statistics for interpretation (*Intellectus Statistics, 2019*; *RStudio-Team, 2020*). For the purposes of performing alpha and beta diversity analyses and visualization of the plots, we used *MicrobiomeAnalyst* and *METAGENassist* which are web-based platforms used for the analysis of microbiota data (*Arndt et al., 2012*; *Dhariwal et al., 2017*). Biodiversity census, either culture-based or culture-independent, require a consensus on diversity metrics. That is, standardized methods for the measurement of biodiversity at different scales exist, such as Chao1 observed species metrics or diversity indexes of Shannon or Simpson, initially used to monitor biodiversity (*Roswell, Dushoff & Winfree, 2021*). Here, we cataloged the grown molds (culture-based approach) and used the species table of abundances to calculate alpha and beta diversity estimates. For data filtering, processing and normalization, we used a low-count filter based on 10% prevalence and no data rarefaction was performed. Normalization was done row-wise, log transformed on the total sum, in *METAGENassist* so that the data could follow a Gaussian distribution and samples could be more comparable to each other. Statistical analyses for alpha diversity, were done using compare_alpha_diversity.py in QIIME which uses a non-parametric *t*-test with Monte Carlo permutations. For Beta-diversity analyses, PCoA, Jensen–Shannon Divergence and permutational MANOVA (PERMANOVA) were used as ordination, distance and statistical methods, respectively. Taxonomic bar plots were created using percentage abundance and a taxa resolution of the top n taxa, with $n = 10$. To identify significant taxa signatures at genus level between two different groups, a Linear Discriminant Analysis (LDA) Effect Size (LEfSe) algorithm for biomarker discovery was used (*Segata et al., 2011*). This method allowed us to identify statistically significant fungal features that differ between the indoor and outdoor environments of homes classified by self-reported degree of water damage, expressed as LDA values shown in boxplots and using either Original (uncorrected) or FDR (False Discovery Rate)-adjusted *p*-value of 0.05. In addition, we compared the fungal profile (alpha, beta, abundance and biomarker analysis) between dry and wet (water-damage + flooded) homes.

## RESULTS

During the first sampling period, a total of 50 homes were sampled (Fig. 1). For the second sampling period, a total of 35 homes completed the follow-up sampling event (70% retention rate). During the first sampling period in 2018, 10 homes were in a non-flooded zone and 40 were in a previously flooded zone as per FEMA's classification scheme following Hurricane María. For the second sampling period in 2019, 8 of 35 homes were in a dry FEMA zone and 27 in a previously flooded area according to FEMA. The self-reported data on flooding conditions provided by participants yielded a more detailed assessment of actual flooding and water damage conditions within the study homes (Table 1). It is worth

**Table 1 Number of homes categorized by self-reported degree of water damage.** The total number of samples for each sampling period, total number of fungal colonies observed, and total number of unique fungal species present are summarized for each category.

| Self-reported degree of water damage | Sampling period | Number of homes | Number of samples | Number of fungal colonies | Number of fungal species |
|---|---|---|---|---|---|
| Dry | First | 13 | 124 | 4,229 | 40 |
| | Second | 11 | 108 | 1,925 | 33 |
| Water- Damage | First | 24 | 240 | 10,624 | 42 |
| | Second | 15 | 150 | 7,136 | 35 |
| Flooded | First | 13 | 121 | 3,606 | 38 |
| | Second | 9 | 90 | 4,893 | 31 |

noting that the FEMA classification does not take into account details that can influence the extent of the flooding (*e.g.*, some homes were on second floors and were not flooded even though they were located in an area categorized by FEMA as flooded). Of the 50 homes sampled in 2018, 13 houses were reported by household members to be dry, 24 had water-damage, and 13 had been subjected to flooding. Of the 35 homes sampled in 2019, 11 houses were reported by household members to be dry, 15 had water-damage, and 9 had been subjected to flooding. All the homes were normally found to be naturally ventilated *via* open doors and windows. Fans and/or an occasional air conditioner were also present in some homes but these were turned off during air sample collection.

No significant differences in median relative humidity levels inside dry, water-damage and flooded homes were observed during the first sampling period (Fig. 2A). In contrast, the indoor environment of flooded homes had significantly higher relative humidity levels during the second sampling period when compared to dry homes ($p$-value $= 0.007$) (Fig. 2B).

Since outdoor fungal spore concentrations are important in the context of tropical homes with natural ventilation, we reviewed these data for the sampling periods of our study. Outdoor levels of fungal spores, reported at the San Juan AAAAI station, were higher during the first sampling period (an average of 86,478 spores/m$^3$ and 103,762 spores/m$^3$ in August and September of 2018, respectively) when compared to the second sampling period (an average of 44,615 spores/m$^3$ and 38,632 spores/m$^3$ in June and July of 2019, respectively). Similar results were obtained for outdoor culturable fungi (CFU/m$^3$) measured outside the study homes with 1.88- and 1.12-times higher levels during the first sampling period when compared to the second sampling period in G25N and MEA, respectively (data not shown, not statistically significant). As outdoor San Juan fungal spore concentrations follow a seasonal pattern related to seasonal precipitation, precipitation levels during the sampling periods were also compared (*Quintero, Rivera-Mariani & Bolaños Rosero, 2010*). Precipitation levels were higher in August 2018 (first sampling period, total monthly precipitation of 6.29″) and in July of 2019 (second sampling period, total monthly precipitation of 6.72″) (*NWSFO, 2014*).

We evaluated the relationship between outdoor and indoor fungal concentration (CFU/m$^3$). For indoor levels we used the culturable measurements median of the indoor

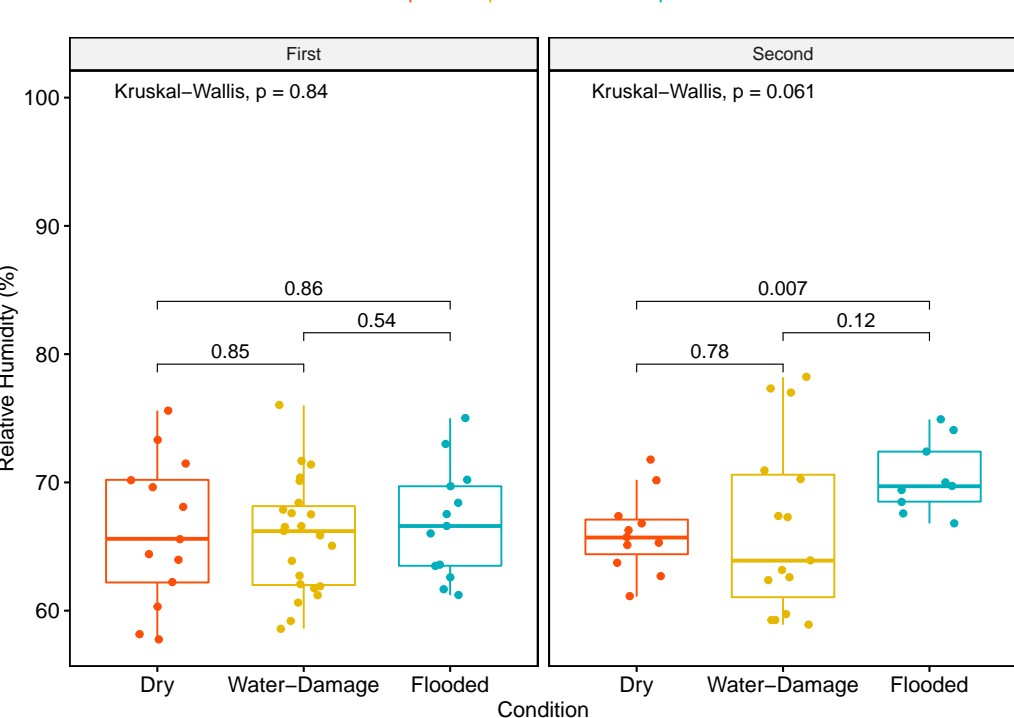

**Figure 2  Indoor relative humidity (%) in homes as a function of self-reported degree of water damage (Dry, Water-Damage, Flooded).** (A) First sampling ($n = 50$ homes). (B) Second sampling ($n = 35$).

locations at each home. For the first sampling period, 12 months after Hurricane María, outdoor and indoor fungal concentration correlated (CFU/m$^3$) ($r = 0.47$, $p < .001$, 95% CI [0.22–0.66]). This correlation indicates that as outdoor fungal colonies increased, indoor fungal colonies also tended to increase. Linear regression model were significant, $F(1,48) = 60.79$, $p < .001$, $R^2 = 0.56$, indicated that outdoor levels explained approximately 56% of the variance in indoor fungal colonies (Fig. 3A). For the second sampling period, 22 months after Hurricane María, a significant positive correlation was still observed between outdoor and indoor fungal colonies ($r = 0.71$, $p < .001$, 95% CI [0.49–0.84]). The correlation coefficient between outdoor and indoor was 0.71, indicating a large effect size. The results of the linear regression model were also significant, $F(1,33) = 11.40$, $p = .002$, $R^2 = 0.26$, with approximately 26% of the variance in indoor fungal colonies explained by outdoor fungal colonies (Fig. 3B).

We evaluated indoor culturable fungi concentrations as a function of self-reported degree of water damage. During the first sampling period in 2018, no significant differences in culturable fungi concentration were observed in both culture media evaluated (Figs. 4A and 4B). For the second sampling period, indoor CFU/m$^3$ increased with degree of water damage for both culture media (Figs. 4A and 4B). Both flooded (Median = 758) and water-damage homes (Median = 408) had 2.7 and 1.4 times higher CFU/m$^3$ when compared to dry non-flooded homes (Median = 283) in the G25N culture media (Fig. 4A), although

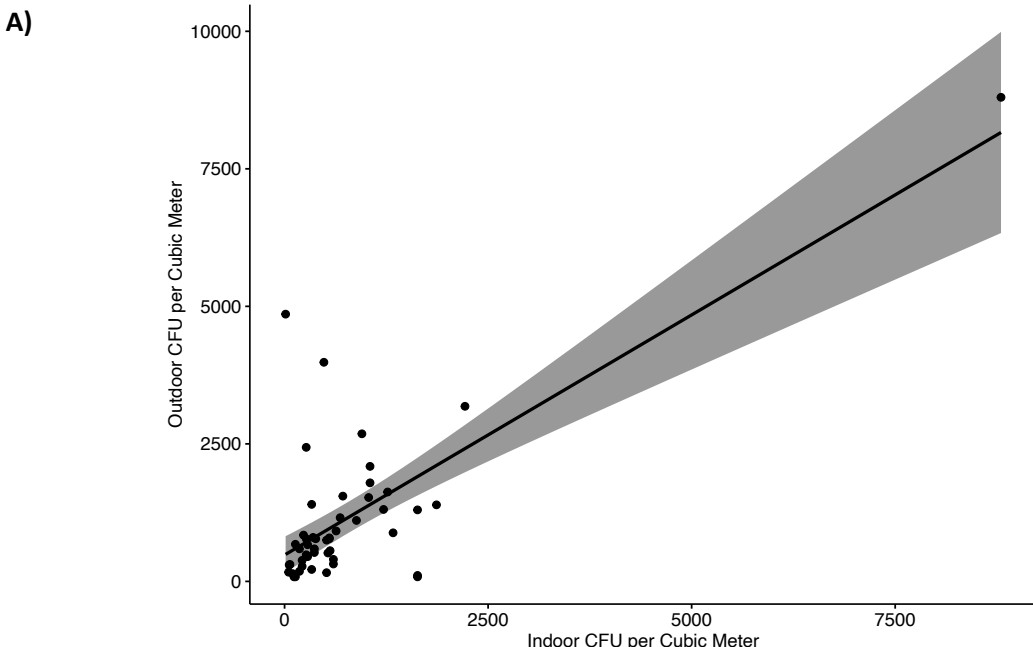

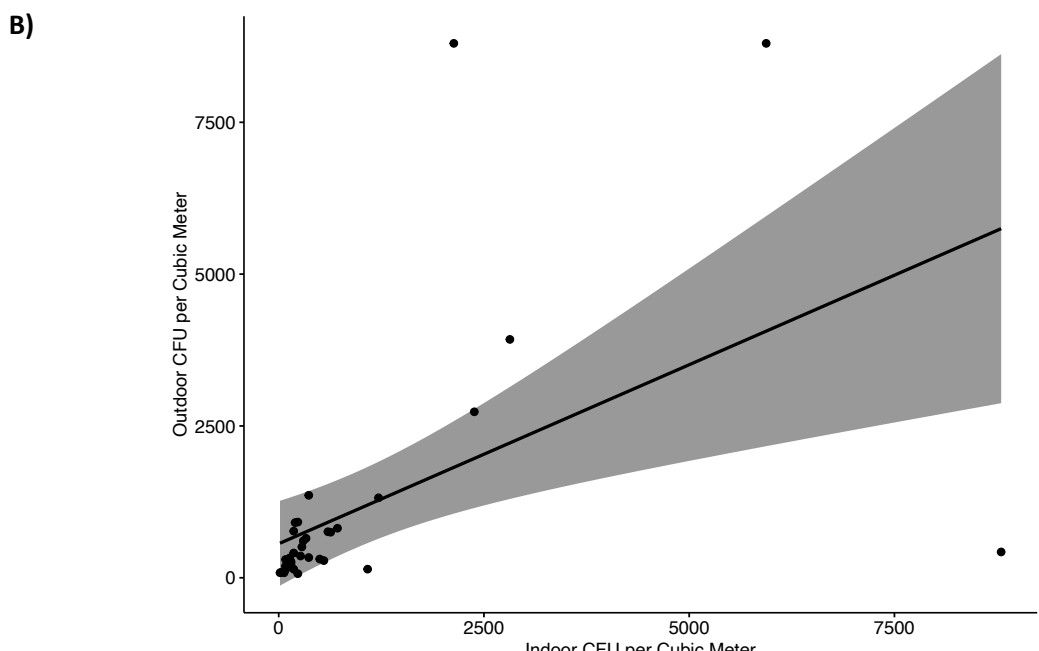

**Figure 3 Linear regression of indoor and outdoor culturable fungi reported as colony forming units (CFU) per cubic meter (m³) of air in G25N culture media.** (A) First sampling (2018) and (B) Second sampling (2019).

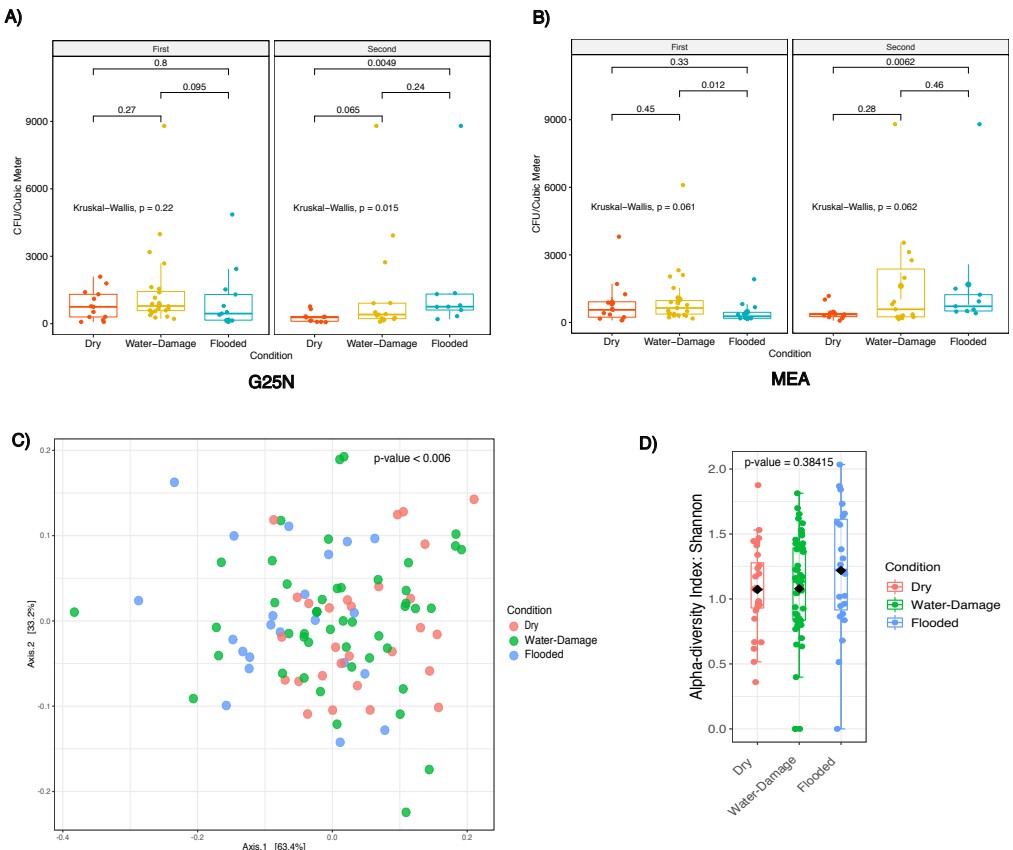

**Figure 4** **Fungal concentration and profile by self-reported degree of water damage (Dry, Water-Damage, Flooded).** (A) G25N culture media. (B) MEA culture media. (C) Beta-diversity analysis for first sampling (2018). (D) Alpha-diversity analysis for first sampling (2018). (A, B) Red, Yellow and Blue boxplots correspond to Dry, Water-Damage and Flooded homes, respectively; first sampling $n = 50$ homes and second sampling $n = 35$ homes; Kruskal-Wallis test, $p$-value $< 0.05$ was considered significant. (C) Principal coordinate analysis (PCoA) performed with Jensen–Shannon Divergence. (D) Chao1 richness estimate.

only statistically significant between flooded and dry homes ($p$-value $< 0.005$). Results from the MEA culture media (Fig. 4B) during the same period showed 2 and 1.6 times higher CFU/m³ in flooded (Median $= 716$) and water-damage homes (Median $= 583$) when compared to dry, non-flooded homes (Median $= 358$), although only statistically significant between flooded and dry homes ($p$-values $< 0.01$).

Beta-diversity analysis using PCoA showed statistical differences ($p$-values $< 0.006$) between fungal communities as a function of self-reported degree of water damage during the first sampling period (Fig. 4C). Alpha-diversity analysis, using the Chao1 richness estimator, showed no significant differences ($p$-value $= 0.38415$) in richness associated with water damage (Fig. 4D, Table S1).

During the first sampling period, the fungal taxa present in the indoor environment of dry homes had similar relative abundance levels to those present in the outdoor environment and were dominated by non-sporulating fungi (Fig. 5A). Water-damage homes were

still dominated by non-sporulating fungi, but there was an increase of *Aspergillus* taxa in the indoor environment. Specifically, during the first sampling period, the indoor environment of flooded homes was enriched by *Aspergillus* species when compared to the outdoor samples. During the second sampling period, enrichment of *Aspergillus* taxa in flooded homes was not observed (Fig. 5B). Both indoor and outdoor environments were dominated by non-sporulating fungi regardless of the degree of water-damage in the homes sampled. Linear discriminant analysis (LDA) together with effect size measurement (LEfSe) method identified several statistically significant fungal features at the genus level (Fig. 5C). Higher LDA scores represent increased abundance of fungal biomarkers in the samples classified by sampling location (indoor *versus* outdoor) and self-reported degree of water damage (dry, water-damage, flooded). During the first sampling period, enrichment of non-sporulating fungi was identified in the outdoor samples of dry homes, whereas *Rhodotorula* was significantly enriched in the outdoor samples of flooded homes (Fig. 5C). In contrast, *Aspergillus* was significantly enriched in the indoor environment of flooded homes during the first sampling period. During the second sampling period no significant features were found.

During the first sampling period, *Aspergillus* spp. and *Penicillium* spp. were more abundant in indoor samples, whereas non-sporulating fungi and *Cladosporium* spp. were more abundant outdoors (Fig. 6A). For the second sampling period, Trichoderma was more abundant indoors while *Aspergillus*, non-sporulating fungi and *Eurotium amstelodami* were more abundant outdoors (Fig. 6B).

Beta-diversity analysis using PCoA showed differences in the fungal communities of dry homes grouped by sampling location (indoor and outdoor) and sampling period (first and second) (Fig. 7C). No differences in group clustering were identified in samples from wet homes (Fig. 7A). Alpha-diversity analysis, using the Chao1 richness estimate, indicated no significant differences ($p$-value = 0.1288) in the number of unique taxa in the dry homes (Fig. 7D, Table S2) and a significant increase in the taxa present in the indoor samples from the second sampling period in wet homes ($p$-value = 0.002) (Fig. 7B). *Aspergillus* spp., *Cladosporium* spp., non-sporulating fungi and *Penicillium* spp. were the most abundant fungi present in the indoor environment of both dry and wet homes during both sampling periods (Fig. 8). Non-sporulating fungi and *Penicillium* spp. abundance inside dry homes decreased whereas *Aspergillus* spp. and *Cladosporium* spp. abundance increased during the second sampling period when compared to the first sampling period (Figs. 8A and 8C). The same trend was observed for wet homes (Figs. 8B and 8D). *Aspergillus* spp. and non-sporulating fungi were identified as biomarkers of dry homes, both with an FDR-adjusted $p$-value = 0.005 (Fig. 9). *Aspergillus* spp. and non-sporulating fungi were more abundant in the indoor and outdoor environments of the second and first sampling periods, respectively (Figs. 9A and 9B).

## DISCUSSION

Fungal quantification and fungal community characterization after Hurricanes are important to develop guidelines for post-flood recovery efforts. In this work, we analyzed

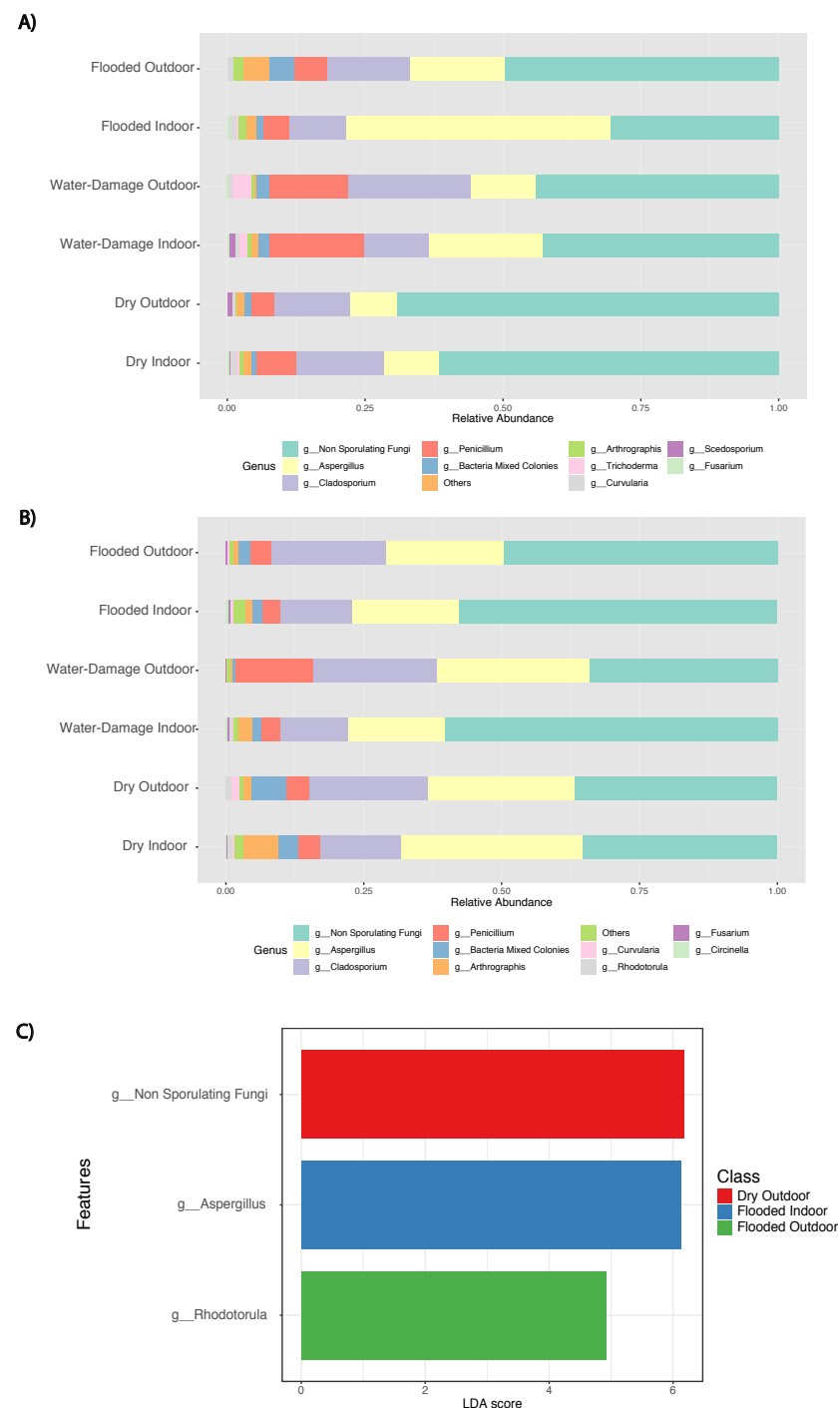

**Figure 5 Abundance taxa bar plots and histogram of Linear Discriminant Analysis (LDA) Effect Size (LEfSe) by sampling location (Indoor, Outdoor), self-reported degree of water-damage (Dry, Water-Damage, Flooded) and sampling period (First, 2018 and Second, 2019).** (A) Abundance taxa bar plots during first sampling (2018). (B) Abundance taxa bar plots during second sampling (2019). (C) LDA during first sampling (2018), FDR-adjusted $p$-value = 0.05. Log LDA score computed for differentially abundant taxa (genus level) with cut-off LDA score > 2.0.

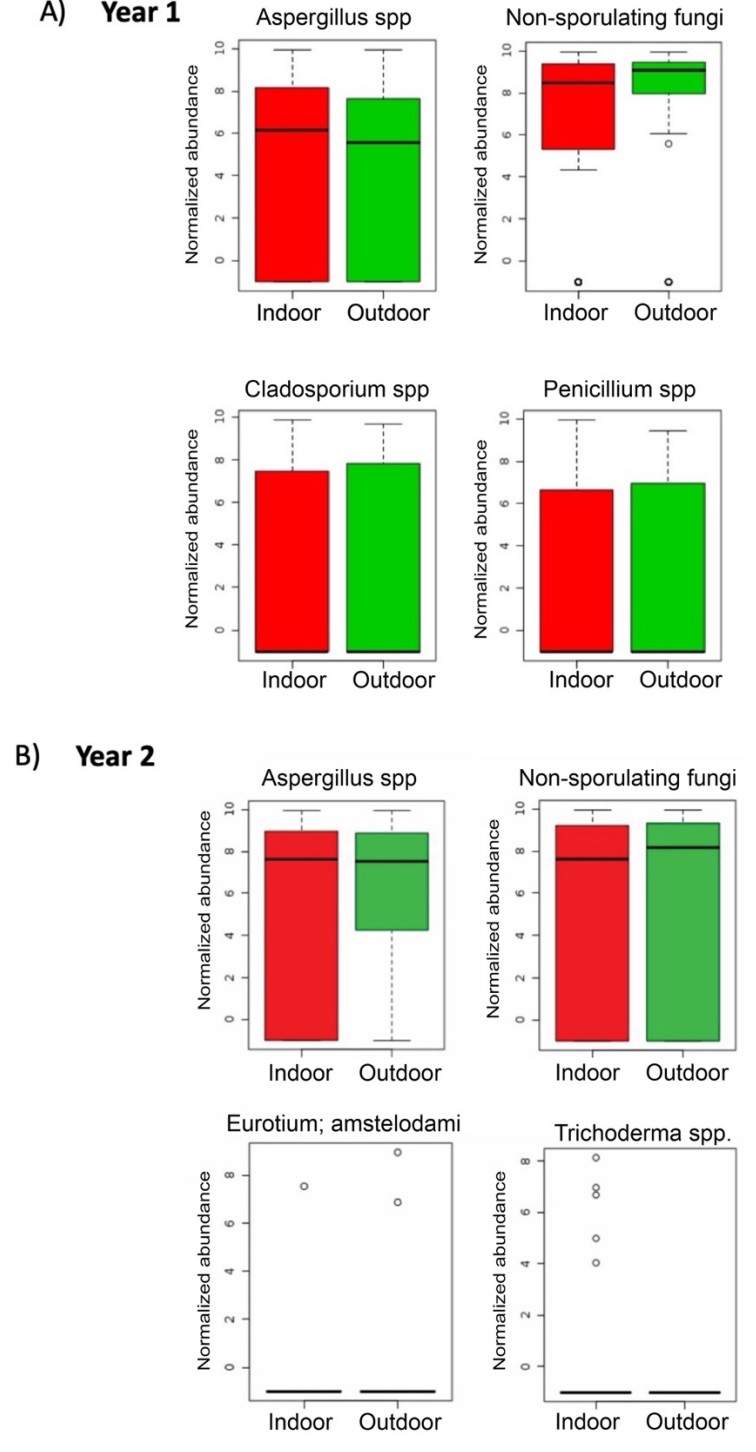

**Figure 6** **Individual fungi normalized abundance boxplots by sampling location (Indoor, Outdoor).**
(A) Individual fungi abundance boxplots during first sampling (2018). (B) Individual fungi abundance boxplots during second sampling (2019). (A, B) Red and Green boxplots corresponds to indoor and outdoor samples, respectively.

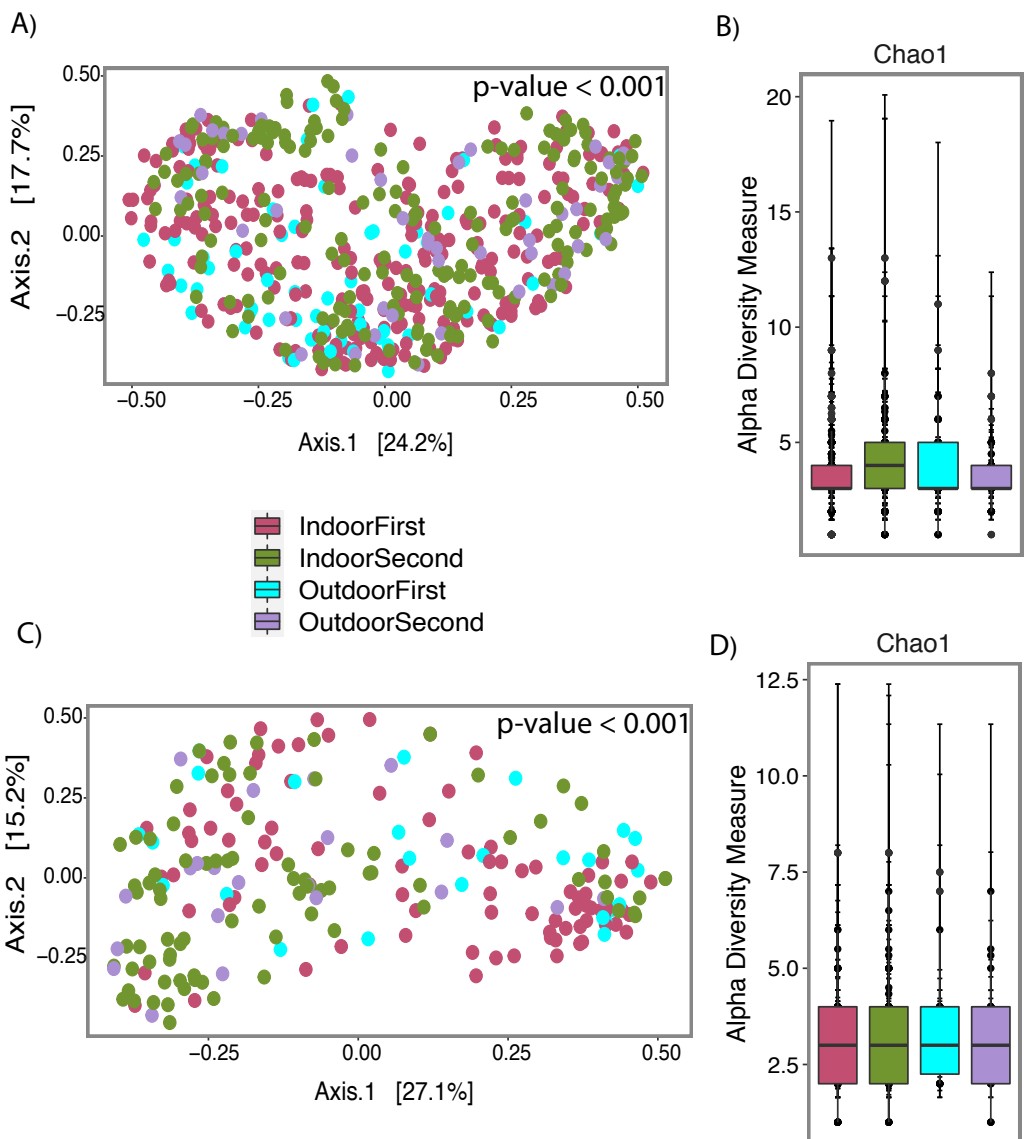

**Figure 7** **Fungal profile of wet (Water-Damage and Flooded homes combined) and dry homes during both sampling periods (First and Second).** (A) Alpha-diversity analysis for wet homes. (B) Alpha-diversity analysis for wet homes. (C) Beta-diversity analysis for dry homes. (D) Alpha-diversity analysis for dry homes. (A, B, C, D) Pink, Green, Light Blue and Purple colors corresponds to indoor samples from first sampling, indoor samples from second sampling, outdoor samples for first sampling, and outdoor samples from second sampling; respectively. (A, C) Principal coordinate analysis (PCoA) performed with Jensen Shannon Divergence. (B, D) Chao 1 richness estimate.

culturable fungal levels in homes located within a neighborhood in San Juan (PR) affected by Hurricane María's extensive flooding. Culturable fungi levels (CFU/m³) were higher

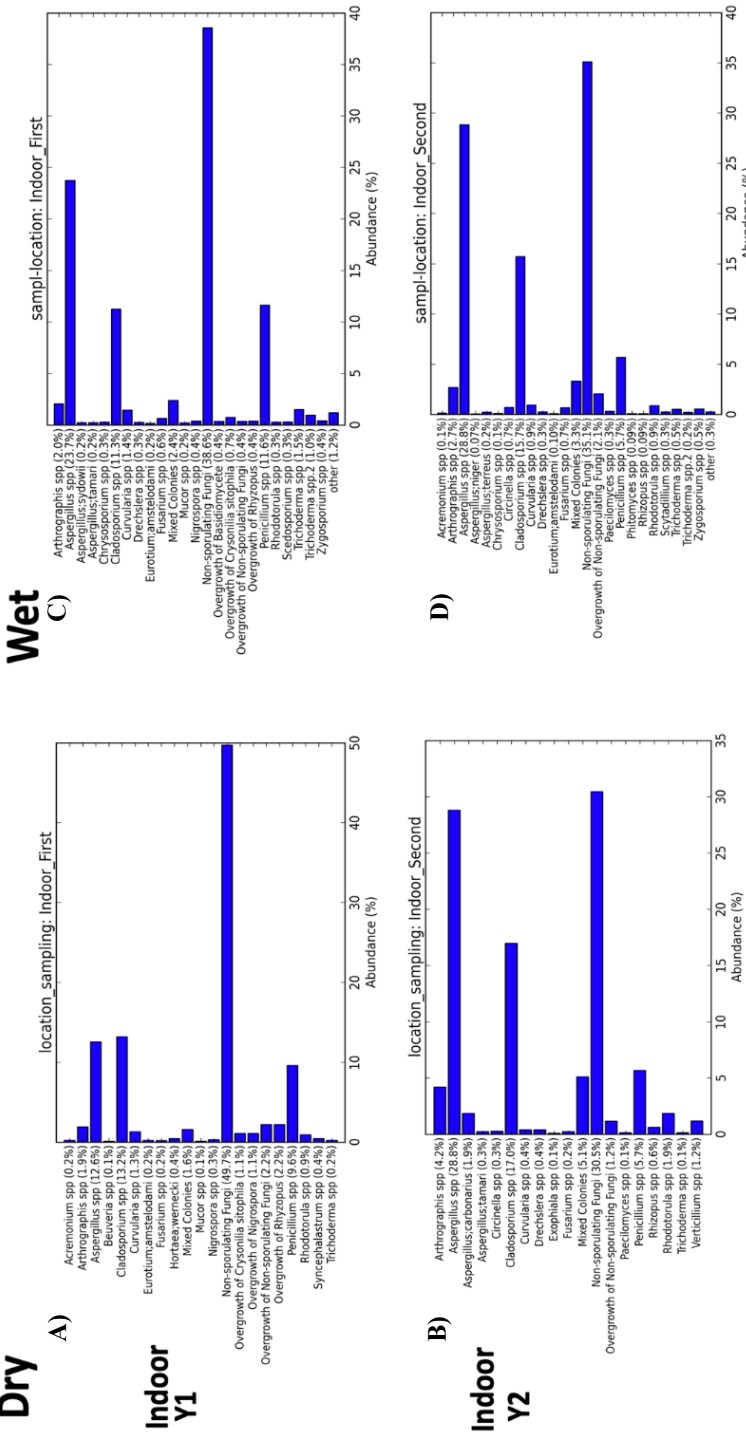

**Figure 8  Indoor fungal isolates abundance percent by sampling period (First, Second).** (A) Samples from indoor environment of dry homes during the first sampling period. (B) Samples from indoor environment of wet homes during the first sampling period. (C) Samples from indoor environment of dry homes during the second sampling period. (D) Samples from indoor environment of wet homes during the second sampling period.

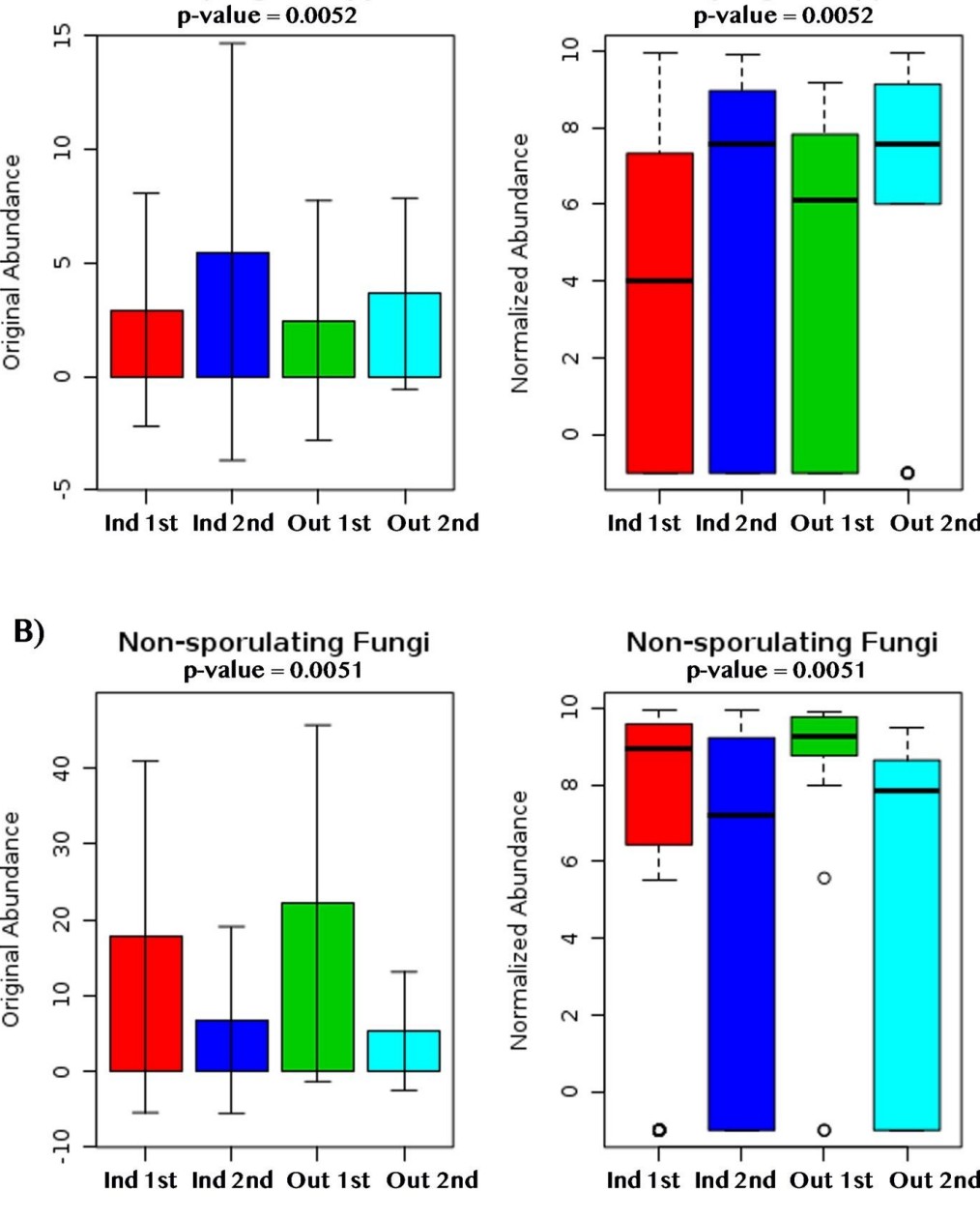

**Figure 9** **Fungal dry biomarkers by sampling location (Indoor, Outdoor) and sampling period (First, Second).** (A) *Aspergillus* spp. abundance levels. (B) Non-sporulating fungi abundance levels. (A, B) Red, Dark Blue, Green and Light Blue boxplots corresponds to indoor samples from first sampling, indoor samples from second sampling, outdoor samples for first sampling, and outdoor samples from second sampling; respectively.

inside previously flooded homes 22 months following Hurricane María as compared to the levels measured inside dry non-flooded homes (Fig. 4B). This result is consistent with previous studies conducted in the aftermath of hurricanes, which have typically found higher fungal growth and fungal components (spores, endotoxins and glucans) inside homes and areas that sustained moderate to severe water-damage and/or flooding (*Chew et al., 2006*; *Riggs et al., 2008*; *Barbeau et al., 2010*). Water damage and flooding may increase humidity and moisture levels inside the affected home, including wetting furniture and household construction materials, and create favorable conditions for fungal growth.

Since humidity contributes to fungal proliferation, we measured relative humidity levels in the homes during the different sampling periods (first and second). Almost two years after Hurricane María we found higher relative humidity levels inside flooded homes; however, this trend was not evident during the first year after Hurricane María (Figs. 2A and 2B). The ground and surrounding areas of flooded homes were inundated, during and after Hurricane María by rain and phreatic water (underground water). Although not measured through in this research, this could have also impacted relative humidity levels in the flooded homes that are also located in shallow water table areas. During the first sampling period, no differences in relative humidity levels were observed possibly due to closeness of the sampling to the Hurricane and flooding events.

We analyzed our culture-based data using different tools typically applied to molecular-based data. These analyses and visualizations provided additional insight into the fungal communities present in our samples as a function of different variables (degree of water damage, indoor *versus* outdoor sampling location). We found significant differences in fungal composition and structure when analyzing beta-diversity by self-reported degree of water exposure during the first sampling period (*p*-value $< 0.001$), meaning that the distribution of the fungal species recovered varied between the categories analyzed (Fig. 4C). However, there were no significant differences in alpha diversity (Fig. 4D). Meaning there were no differences in the number of species within the categories studied. Results from abundance taxonomic bar plots highlighted the shift from fungal communities present in the outdoor environment (typically non-sporulating fungi) to enrichment of taxa previously associated with damp environments and human disease (Figs. 5A and 5B). Non-sporulating fungi refer to fungi that fail to produce spores on culture media, therefore identification through microscopy techniques is not possible. Non-sporulating fungi are most likely members of the Basidiomycota phylum, which compose approximately 60% of the fungal spores present in the outdoor air of PR (*Rivera-Mariani et al., 2020*; *AAAAI, 2021*). Thus, the indoor environment of dry non-flooded homes appears to reflect the outdoor environment (Fig. 5A). This result was expected, since the study homes, as with most homes in PR, are naturally ventilated allowing for fungal spores present in the outdoor air to enter the homes through doors and windows. In contrast, water-damage homes yielded an increase in abundance of *Aspergillus* taxa which eventually dominated the indoor environment of flooded homes (Figs. 5A and 5C). Approximately 180 species of *Aspergillus* exist, with *A. fumigatus, A. terreus, A. niger, A. flavus* being the most frequent causative agents of human fungal infection, including aspergillosis, asthma and allergic rhinitis (*Sugui et al., 2014*; *Seyedmousavi et al., 2018*). Moreover, *Aspergillus* taxa are associated

with damp environments (*Miller, Haisley & Reinhardt, 2000*; *Chew et al., 2006*; *Solomon et al., 2006*; *Rao et al., 2007*; *Schwab et al., 2007*; *Riggs et al., 2008*; *Bolaños-Rosero et al., 2013*; *Flores et al., 2013*). The shift in fungal taxa from the first to the second sampling period in water-damage and flooded homes (from *Aspergillus* species to non-sporulating fungi predominance) indicates that 22 months after Hurricane María, there has been a recovery in the fungal communities to the taxa normally present in the tropical environment of Puerto Rico (Fig. 5B). *Rhodotorula* was found significantly enriched in the outdoor environment of flooded homes (Fig. 5C). This unicellular red-orange pigmented yeast is part of the Basidiomycota division and has been previously associated with water-damaged materials and moisture damage in buildings (*Andersen et al., 2011*; *Adams et al., 2020*) .

Finally, the most abundant fungi recovered during the first sampling period from both the indoor and outdoor environments of dry and wet homes were *Aspergillus*, *Cladosporium*, *Penicillium* and non-sporulating fungi. The first three fungi correspond with findings of other studies, while non-sporulating fungi represent the characteristic fungi from the tropical outdoor environment of PR (Figs. 6A, 8 and 9) (*Miller, Haisley & Reinhardt, 2000*; *Chew et al., 2006*; *Solomon et al., 2006*; *Rao et al., 2007*; *Schwab et al., 2007*; *Riggs et al., 2008*; *Bolaños-Rosero et al., 2013*; *Flores et al., 2013*).

Our study provides a unique and valuable data set. First, we evaluated fungal concentration and composition during two sampling periods one and two years in the aftermath of hurricanes, María. Even one year after Hurricane María, the homes sampled had differences in fungal communities according to the degree of water damage. Almost 2 years after the hurricane, we found that fungal communities indoor began to recover and resemble the outdoor fungal communities. Most of the studies that evaluated fungal growth and spore levels after a hurricane were carried out within 6 months after the natural disaster. Second, we collected our data through culture-based methods which allows analysis of viable fungi present in the sampled environment. We acknowledge the limitations of this approach for fungal identification, however useful information was still gained regarding indoor fungal amplification and diversity. Third, we used two different culture media for the recovery of fungal isolates, MEA and G25N allowing for the characterization of a wide range of fungi from hydrophilic to xerophilic fungi, respectively. Our results showed that G25N is more useful for the collection of fungal isolates that usually amplify (indoor concentrations greater than the outdoors) under damped or flooded conditions. Since G25N allows slow and discrete colony growth, all fungi can grow without a dominating genus overgrowth occupying the culture plate surface as it happens in MEA. However, the use of both culture media is recommended to recover as many fungal species as possible. We also considered the levels of outdoor fungal spores reported by the San Juan AAAAI station, which were highest during the first sampling period and contributed to the indoor fungal concentration, agreeing with an outdoor-indoor air continuum. Also, the fungal isolates obtained from these samples agreed environmental fungi from PR obtained through the San Juan AAAAI station. Lastly, we applied tools originally develop for molecular-based data for the analysis of our culture data gaining additional insights into fungal communities' composition, richness, abundance and biomarkers significantly enriched in our samples. By applying tools originally develop for data generated by molecular biology approaches

for the analysis of culture-based data, we identified fungal features characteristic between dry and flooded homes. Particularly, we observed a significant shift from *Aspergillus* taxa to non-sporulating fungi inside flooded homes between the first and second sampling period.

In summary, this data highlights dynamic changes in indoor fungal populations, characterized by different types of fungi between dry and flooded homes. More specifically, dry homes were dominated by fungi present in the outdoor environment highlighting the naturally ventilated types of homes common in PR. On the other hand, flooded homes presented amplification of *Aspergillus* species. Additionally, our study results indicate that in naturally ventilated homes, the outdoor fungal levels have a significant effect on indoor fungal levels. This brings attention to the outdoor environment. Thus, after-flood cleanup and remediation at a community scale may help address this issue. Since tropical homes are characterized by an outdoor-indoor air continuum (Figs. 3A and 3B), it is extremely important to maintain clean outdoor surroundings after the disaster. In this study, we observed high quantities of water-damage materials, wood and trees debris outside many of the homes sampled. These organic materials could have also contributed to outdoor filamentous fungi proliferation that eventually entered the homes *via* open windows and/or doors. To apply these suggestions in the context of low-income communities, assistance from local government is warranted. Disaster relief for flooding victims, including debris removal from the surroundings, covered-expenses home repairs, materials and/or furniture replacement, providing cleaning materials and basic education on flood preparedness and recovery. Furthermore, to prevent indoor fungal proliferation, it's recommended to maintain a clean indoor environment after flooding events. Building problems should be repaired and water-damaged materials must be cleaned, dried and/or removed (*Institute of Medicine, 2004*). Remediation of mold-damaged materials, through cleaning or replacement, decreases or eliminates mold growth (*Solomon et al., 2006*; *Riggs et al., 2008*; *Jayaprakash et al., 2017*). It is important to identify indoor moisture sources such as roof, walls and plumbing leaks, condensation, floods and high relative humidity. An additional recommendation is to keep indoor relative humidity levels below 60%, ideally between 30% and 50%, in order to inhibit mold growth (*EPA, 2021*). Moreover, dehumidifiers and High-Efficiency Particulate Air (HEPA) filters can help control fungal growth and spore levels, respectively (*Mazur & Kim, 2006*).

## ACKNOWLEDGEMENTS

The authors want to thank all the families participating in this research. Additionally, we thank the team of research students that participated in this project. This research project is in partial fulfilment of Lorraine N. Vélez-Torres doctoral thesis dissertation.

### Funding

This work was supported by a grant from the National Institutes of Health (NIH) number R21 ES029762-0101 and partial funds were received from MBRS-RISE program of UPR-MSC (award number R25GM061838), PR-INBRE BiRC NIH/NIGMS P20 GM103475 and

NIMHD CCRHD grant number U54 MD007600. The funders had no role in study design, data collection and analysis, decision to publish, or preparation of the manuscript.

## Grant Disclosures

The following grant information was disclosed by the authors:
The National Institutes of Health (NIH): R21 ES029762-0101.
MBRS-RISE program of UPR-MSC: R25GM061838.
NIMHD U54 MD007600 and NIGMS: P20GM103475.

## Competing Interests

The authors declare there are no competing interests.

## Author Contributions

- Lorraine N. Vélez-Torres performed the experiments, analyzed the data, prepared figures and/or tables, authored or reviewed drafts of the paper, and approved the final draft.
- Benjamín Bolaños-Rosero conceived and designed the experiments, performed the experiments, authored or reviewed drafts of the paper, and approved the final draft.
- Filipa Godoy-Vitorino analyzed the data, prepared figures and/or tables, authored or reviewed drafts of the paper, and approved the final draft.
- Felix E. Rivera-Mariani analyzed the data, authored or reviewed drafts of the paper, and approved the final draft.
- Juan P. Maestre, Kerry Kinney and Humberto Cavallin conceived and designed the experiments, authored or reviewed drafts of the paper, and approved the final draft.

## Ethics

The following information was supplied relating to ethical approvals (i.e., approving body and any reference numbers):

IRB UPR Rio Piedras.

## Data Availability

The raw data are available in the Supplemental File.

## Supplemental Information

Supplemental information for this article can be found online at http://dx.doi.org/10.7717/peerj.12730#supplemental-information.

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
