# Peer review of "Hurricane María drives increased indoor proliferation of filamentous fungi in San Juan, Puerto Rico: a two-year culture-based approach"

_PeerJ, doi:10.7717/peerj.12730_

## Round 0.1 · original submission · Major Revisions

The manuscript received mixed reviews, from minor to reject. The topic is interesting and most of the revisions suggested are minor but I share the concern that the method of fungi identification is inappropriate. Molecular techniques (Sanger sequencing of PCR-amplicon, MALDI-TOF …), rather than morphological classification, are the current standard methods for microbial identification. Indeed the paper cited when the identification method was presented (line 205, I presume is Guarro 1999 Clin Micro Rev), emphasized the importance of ribosomal gene sequencing. That was published two decades ago.

To start, this method must be justified in the Introduction and discussed in the Discussion but I am not sure that morphological classification can be justified without more data. It would strengthen a resubmission if a subset of the fungi isolates were classified with both morphological and molecular methods. If the isolates are no longer available, this could involve collecting another library.

Reviewer 1 ·

Basic reporting

This paper reported about airborne fungi in flooded and non-flooded homes in Puerto Rico after a hurricane. The study used culture techniques to quantify and identify fungi collected by a multiple-jet impactor. Fungal concentration, composition, and diversity were analyzed and compared between flooded and non-flooded homes. I agree that the topic is important. The manuscript is well-written with professional structure. Relevant references are also properly cited.

Experimental design

The goal of the study is clear, i.e., to investigate mold problems in Puerto Rico, the less-studied region vulnerable to tropical weather. I agree that the findings are novel. The methods used seem appropriate with sufficient details provided in the manuscript, though I have some reservation about statistical methods used in this study (see my comments in other section).

Validity of the findings

I agree that the study is novel. I also agree that the findings of this study is useful in this field of research. The sampling and analysis was repeated with the sufficient number for meaningful replication. I have some comments on the statistical methods used, however. My comments follows:

Statistical analyses used in this study don't seem to be appropriate. The study sampled ~50 homes, each at 4 indoor locations of living room, kitchen, bedroom and bathroom (Line 177-178). This means that samples are stratified, with “home” as the primary stratum and “location” and “year” as the sub strata. The problem is that each sample is considered independent regardless of sampled “home” and “location”. For instance, the samples collected at different locations in the same home are likely not independent (since they were collected in the same home). Looking at Figures 2 and 4 (also other figures showing statistical analyses), for example, it can be misleading as if the study sampled at ~200 homes. Some of the results are collected in the same home, so it may be more appropriate to consider them as replicates. To me, it is more reasonable to calculate the average of 4 indoor locations of each home, and then to compare the calculated averages between flooded and non-flooded homes. Alternatively, stratified analyses should be performed since the results are stratified by "home", "location", and “year”.

Additional comments

My additional comments follows”

Genus-level analysis
The authors said that analyses were at the genus level (Line 199). However, some taxa (e.g., Aspergillus) were reported at the species level, e.g., Aspergillus terreus, Aspergillus carbonarius, Aspergillus tamari, in Figure 5. The authors also used the term “Aspergillus spp.”. Does the term “Aspergillus spp.” or “Aspergillus” include or exclude the specific Aspergillus species, such as Aspergillus terreus, Aspergillus carbonarius, Aspergillus tamari? Were they separately analyzed from “Aspergillus spp.” for statistical analyses? It is confusing since the species-level information are included in the genus-level information.

Line 50 and others
The phrase "using molecular tools" is misleading since it gives impression that this study used molecular tools. More accurately, this study used tools originally developed for the data generated by molecular biology approaches.

Line 180
When was the number of open and closed windows counted?

Line 181
Usually, MEA stands for malt (not maltose) extract agar.

Line 190
Does this software have function of positive-hole correction? The positive-correction is necessary to correct for deposition of fungal spores on the same spot of multi-jet impactor.

Line 285
Why were only living room samples selected?

Line 349 and others
The word “spp.” shouldn’t be italic.

Line 414
division --> phylum

Lines 456-457
I disagree with this sentence. Now, DNA is used as a basis of fungal taxonomy. Therefore, all of scientifically proven species should have the molecular information. If the molecular information doesn't exist, it means that that species is novel. If the species is novel, we (who are not taxonomists) cannot identify regardless of the methods used. We have to ask taxonomists to define it.

Fig. 1
This map is not informative. For instance, there is no scale. Also, it is impossible to know the relative location of the sampling area in the island. Could you re-create?

Reviewer 2 ·

Basic reporting

This is an interesting study where the authors compared culturable fungal concentrations in dry, water-damaged, and flood-damaged homes (total 50) followed by Hurricane Maria in Puerto Rico. The authors may address following limitations of the study design to improve their manuscript.

Experimental design

Lines 132 - 133: This hypothesis is not so significant because it’s already well known that water damage from floods can increase mold growth and consequent higher indoor mold spore levels. Many articles were previously published on this topic. The authors may consider comparing fungal diversity and abundance in flooded versus non-flooded homes and hypothesize a particular type of difference anticipated in the tropical climate of Puerto Rico.
Lines 151 – 152: The sample sizes for different categories of houses were too small. Please acknowledge this limitation and provide some rationale defending this.
Line 170 – 171: One major disadvantage of the air sampling approach was it was solely focused on culturable fungi and many environmental fungi may not be culturable in the traditional MEA media. Different fungal species may proliferate at different rates on the culture media used, which is another problem. The authors used the word ‘spores’ in many places but some hyphal fragments can develop a colony and colony counts may not be equal to spore counts. Therefore, data obtained are not showing the true airborne fungal diversity in the studied homes. Calculated diversity based on culturable approach may not be useful for the understanding health risks from these fungi because allergic risks depend on both culturable and non-culturable fungal exposures.
Lines 205 and 562: This reference is incomplete. How this article enabled identifying so many different types of fungal species is unclear. This is another problem in the experimental design.
Line 266: How the indoor humidity levels became higher in June – July? Was it an effect of outdoor climatic conditions? Please provide an explanation.
Lines 440-441: These fungal species are common everywhere. This is unclear what different types of taxa were identified in flood-affected homes.

Validity of the findings

Impact and novelty not assessed. Meaningful replication encouraged where rationale & benefit to literature is clearly stated.

Additional comments

None

Reviewer 3 ·

Basic reporting

This is an interesting assessment of a culturable fungi after a major hurricane. The convenience sample is a limitation in their paper, but there is still knowledge gained about fungal diversity within this convenience sample.

Experimental design

Three major issues need to be addressed:

1. The authors keep calling their measurements, fungal spores. However, they measured colony-forming units. They did not measure individual spores.

2. Please confirm that the control homes that were considered “dry” had not been water-damaged prior to Hurricane Maria. Prior water damage would make a home an imperfect control.

3. Positive hole correction is necessary for impactor devices. The concept of multiple impactions through holes has been well established since the Andersen impactor was first described in 1958.

Positive hole correction is a recommended method in the American Conference of Governmental Industrial Hygienists (ACGIH):

Macher, J., (ed.) Bioaerosols: Assessment and Control, ACGIH, 1999

Macher JM. Positive-hole correction of multiple-jet impactors for collecting viable microorganisms. Am Ind Hyg Assoc J. 1989 Nov;50(11):561-8. doi: 10.1080/15298668991375164. PMID: 2688387.
https://pubmed.ncbi.nlm.nih.gov/2688387/

In the owner’s manual for the device that the authors used, there is a description of how the positive hole correction should be calculated: https://www.manualslib.com/manual/2034219/Cantium-Scientific-Microbio-Mb2.html?page=30#manual

Validity of the findings

Specific Comments

1. Line 87: Add the word, fragments. Correct the units. “Fungal spores and fragments less than 2.5 µm…”
2. Line 88: Why mention pollen in this paper about fungi? I suggest adding that large fungal spores and clusters of small fungal spores can deposit in the upper airways. Also, allergic rhinitis can be caused by not only upper airway, but lower airway deposition too.
3. Line 108-111: Aspergillus is not the most frequent causative agent of asthma and rhinitis. It is “among the most frequent” triggers of asthma and allergic rhinitis.
4. Line 116-117: Please provide reference for 2017 outdoor fungal spore season being lower. Also, I would change “halted” to just lower levels. I am sure there were still outdoor fungal spores in the air.
5. Line 499-500: Recommended levels of relative humidity to avoid mold growth is lower than 65%. https://www.epa.gov/mold/brief-guide-mold-moisture-and-your-home “Indoor relative humidity (RH) should be kept below 60 percent -- ideally between 30 percent and 50 percent, if possible.”

Additional comments

No comment.

---

## Round 0.2 · accepted · Accept

I made a number of edits for style and format that should be considered in production. Importantly, the references are not formatted consistently. Also, I think you could delete a lot of the text to improve the flow.

Reviewer 1 ·

Basic reporting

I agree that the manuscript is well-written with professional structure. Relevant references are also properly cited.

Experimental design

I agree that the methods used are appropriate with sufficient details provided in the manuscript.

Validity of the findings

I agree that the study is novel. I also agree that the findings of this study is useful in this field of research. I also thank the authors to seriously consider my comment on statistical analysis.

Additional comments

I found that the authors carefully address my comments. I thank the authors for their efforts.

Reviewer 3 ·

Basic reporting

Revised version is well done.

Experimental design

Revised version is well done.

Validity of the findings

Revised version is well done.

Additional comments

No additional comments.